# METRIC LEARNING FOR DETECTION OF LARGE LANGUAGE MODEL GENERATED TEXTS

## ABSTRACT

More efforts are being put into improving Large Language Models' (LLM) capabilities than into dealing with their implications. Current LLMs are able to generate texts that are seemingly indistinguishable from those written by human experts. While offering great quality of life, such breakthroughs also pose new challenges in education, science, and a multitude of other areas. To add up, current approaches in LLM text detection are either computationally expensive or need accesses to the LLMs' internal computations, both of which hinder their public accessibility. With such motivation, this paper presents a new paradigm of metric-based detection for LLM-generated texts that is able to balance among computational costs, accessibility, and performances. Specifically, the detection is performed through evaluating the similarity between a given text to an equivalent example generated by LLMs and through that determining the former's origination. In terms of architecture, the detection framework includes a text embedding model and a metric model. Currently, the embedding component is a pretrained language model. We focus on designing the metric component which is trained with triplets of same-context instances to signify distances between human responses and LLM ones while reducing that among LLM texts. Additionally, we develop and publish four datasets totalling over 85,000 prompts and triplets of responses in which one from human and two from GPT-3.5 TURBO for benchmarking and uses by the public. Experiment studies show that our best architectures maintain F1 scores in between 0.87 to 0.95 across the tested corpora in both same-corpus and out-of-corpus settings, either with or without paraphrasing.

## 1 INTRODUCTION

The advancement in computing technologies has enabled deep neural networks (LeCun et al., 2015) to grow enormous in scales, attributed to the breakthrough of the so-called large language models (LLMs) (Zhao et al., 2023). LLMs are typically deep neural networks that consist of hundreds of millions to hundreds of billions of parameters. Examples of recent LLMs are GPT-3 (Brown et al., 2020) at 175 billion parameters, PaML (Chowdhery et al., 2022), 540 billion parameters, and GPT-4, (OpenAI, 2023) 170 trillion parameters. These gigantic scales provide LLMs with the capabilities of generating significantly high quality expert texts in any domains. Recently, LLMs have become highly accessible to the public with ChatGPT (OpenAI, 2022) using GPT-3.5 TURBO which is renowned for its ease of use and high quality responses in terms of correctness and writing.

ChatGPT-like services have tremendous potentials in aiding societies and improve quality-of-life in a multitude of areas. However, LLMs also come with major challenges. Due to their high levels of sophistication and qualities, it is increasingly more difficult to determine if texts are written by human or LLMs. This poses a major issue in education, science, and any areas that need original writing contents. Coupling with this issue, the literature in detecting synthetic texts seems lagging behind new generative models. Up until mid-2022, studies in detecting LLM-generated texts were relatively limited (Guerrero & Alsmadi, 2022) and rely on training or finetuning computationally expensive supervised classifiers. Furthermore, they were before the breakthrough of major LLMs like GPT or PaML, and therefore the adaptability of such technologies is questionable. The more recent line of synthetic text detection algorithms like DetectGPT (Mitchell et al., 2023) or watermarking (Kirchenbauer et al., 2023) needs access to the LLMs' internal computations, which is not always available to the public and hinder their accessibility.

With such motivation, in this paper, ***we propose a metric-based approach for LLM content detection that is balanced among computational costs, accessibility, and performances***. Due to the generation technologies, LLMs tend to produce similar phrases in texts when receiving the same contexts. Exploiting this, the detection methods will rely on *comparing a given text to an AI-generated reference from LLMs*. Having the assistance from the generative models, a large part of the computational burdens will be relieved from the detection models. In terms of architectures, *the detection framework consists of a pretrained embedding language model and an empirically-designed deep metric network*. The metric network is trained to signify the similarity between LLM responses while decreasing that between LLM and human responses of with a ***same-context triplet training*** algorithm adapted from the triplet loss function (Schroff et al., 2015). During the decision making phase, the context of a given text is first prompted to a LLM to obtain a LLM-reference. The text and the LLM-equivalence are then fed to a metric framework to obtain their similarity metric. Finally, the metric is compared against a selected threshold to determine the text origination.

To benchmark the proposed models, we further develop four text datasets in the form of context - triplets of responses (one from human and two from GPT 3.5-TURBO). Contexts and human responses are extracted from the Natural Questions (NQ) dataset (Kwiatkowski et al., 2019), Stanford Question-Answering Database (SQUAD) (Rajpurkar et al., 2016), Scientific Questions (SciQ) dataset (Johannes Welbl, 2017), and Wikipedia Scientific Glossary (Wiki) (Wikipedia, 2017). Each context is then used to requested two independent responses from ChatGPT to form the triplet instances. By the end, we obtain $59,945$ entries from the NQ data, $18,813$ from the SQUAD data, $4,419$, SciQ data, and $2,071$, the Wiki data. Experiment studies show that our beset architectures maintain F1 scores mostly in between $0.87$ to $0.95$ across the tested corpora in both same-corpus and out-of-corpus settings, either with or without paraphrasing.

The current scope of our work is on short responses, i.e., within a paragraph averaging five sentences but no less than three. We are also focusing on training the metric network and keep the pretrained embedding model frozen. To sum up, our **contributions** are as follows.

1. An metric-based approach that detects LLM responses for known contexts. Unlike others in the current literature, our method is more light-weighted and does not require access to any LLMs' internal computations. Instead, a response is compared to a LLM-generated one of the same context. The similarity of the two responses decides if the former was written by LLM or a human.

2. Empirically designed end-to-end deep architectures that transform text data into embedding vectors of which distances between LLM texts are minimized and that between LLM and human-written texts are maximized. The architecture is trained using a same-context triplet algorithm adapted to the problem of detecting LLM texts.

3. Four text datasets totaling over 85,000 instances of contexts and triplet of responses in topics ranging from daily lives to sciences for benchmarking. All datasets will be available for the community after the publication of this paper.

The rest of the paper is organized as follows. Section §2 discusses the studies related to our work. In Section §3, we present the developed methodologies in details, including the detection algorithm, definite triplet training, and the framework architecture. Our experiment study is discussed in Section §5. Finally, we conclude our paper in Section §6.

## 2 RELATED WORKS

Up until the middle of 2022, detecting AI generated texts is not a very active research area (Guerrero & Alsmadi, 2022). The typical works in this period attempt to use supervised detection models on top of deep embedding representations (Bakhtin et al., 2019; Solaiman et al., 2019; Ippolito et al., 2019; Fagni et al., 2021). There are two drawbacks to these types of models. First, due to the complexity of text data, supervised classifiers tend to require a large number of parameters to perform well, making them very computationally expensive or difficult to train. In our experiments, a deep classifier of 20 layers and over 12 million parameters could not converge when using MPNet embeddings (Song et al., 2020) as inputs. Additionally, finetuning pretrained LLMs takes hours per epoch, while *our approach takes tens of seconds to below three minutes*. Second, these works were tested before the emergence of multi-billions of parameters LLMs that ChatGPT used. This make their adaptability to data from such models questionable. Furthermore, as the synthetic texts become more sophisticated than ever, the first issue is amplified and more difficult to address.

Another branch to verify the origination of texts is to assess the probabilities of words in texts being generated by a given model (Solaiman et al., 2019; Mitchell et al., 2023). The most recent work in this direction is DetectGPT (Mitchell et al., 2023) which has obtained very good detection accuracy. These approaches, however, are hindered by the fact that they require accesses to the underlying output distributions of the generative language models. In closed or commercial LLMs, such distributions either require purchases or are not available at all. Ultimately, this issue also makes it difficult for the general community to utilize these technologies.

Watermarking is the third method to detect AI-synthesized texts (Kirchenbauer et al., 2023). This method mainly involves modifying a language model so that it injects signature patterns into the output texts. The signatures are indistinguishable by human, but make it much easier for a detection model to recognize traces of AIs. While being proved to be highly accurate, the assumption that a LLM in the wild has been implemented with any watermark mechanisms is fairly strong and cannot be guaranteed or enforced.

Addressing these gaps in the related literature, the proposed framework will be without the needs of expensive resources or accesses to internal computations of the generative LLMs. Instead, decisions are made by comparing the given text to an equivalent sample that is known to be from LLMs. This approach partially shifts the computation burdens, specifically, in generating a LLM reference, to the generative LLMs. The detection model will only focus on contrasting a given text and the LLM reference to determine the former's origination. Accordingly, this family of algorithms will be able to maintain a lower complexity and computational demands while keeping good accuracy rates. ***To our knowledge, this is the first work that utilizes metric learning in detecting LLM-synthesized contents***. Our experiments show that the model can be trained very quickly, a few minutes per epoch, on a corpus of approximately 60,000 questions and triplets of responses, and adapt well to paraphrased data as well as out-of-corpus data with $F1$ in between $[0.87, 0.95]$.

## 3 METHODOLOGIES

Texts that are generated by LLMs tend to be repetitive to some degrees when coming from the similar initial context. This observation has inspired the authors to explore detection frameworks that compare a given text to a LLM one from an equivalent context. In this section, we describe our methodology in details, including the justification for the method, the triplet sampling strategy, and the complete detection architecture.

### 3.1 JUSTIFICATION

Current LLM-based assistants generate contents starting from a *context* which can be a question, a query, or a conversation starter. Due to the probabilistic models that LLMs use to create their contents, the same context will result in repeated patterns in the generated texts. More specifically, key technologies such as *autoregressive generation* (Graves, 2013; Sutskever et al., 2014), *beam search* (Wiseman & Rush, 2016), and *next-token sampling* (Fan et al., 2018; Holtzman et al., 2019), focus on exploring and selecting outputs based on probabilities of each tokens in the vocabulary becoming the next for the current sequence. As the next-token probabilities are computed using a given context then autoregressed, the same contexts will result in similar pools of tokens for selection which leads to repetitions of patterns. This phenomenon has also been observed previously (Welleck et al., 2019; 2020). As an illustration, Figure 1(a) shows an example of responses from GPT-3.5 TURBO for the prompt *"briefly explain machine learning within three sentences"* in four separate sessions; with similar phrases across the responses being bold and highlighted with the same colors. We can observe that there are multiple patterns that get repeated across the four responses. Furthermore, besides the highlighted phrases, there is also a strong resemblance in the flows of ideas and writing styles in the texts.

With this observation, the detection framework will base on the core mechanism of learning similarities among LLM responses and dissimilarities between LLM and human responses. To make decision, a given text is compared to a LLM-generated one for similar contexts. If the similarity is over a selected threshold, the text is classified as originated from LLM, otherwise, a person. This process is illustrated in Figure 1(b).

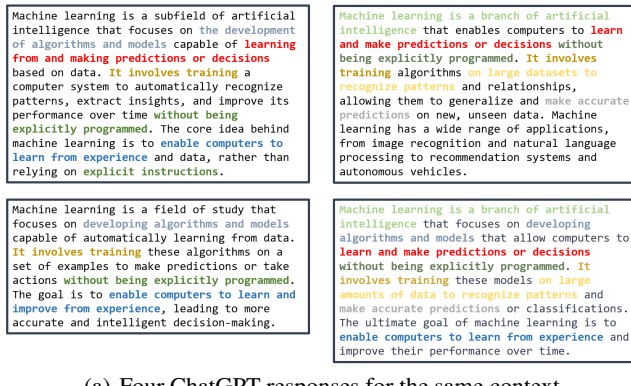

(a) Four ChatGPT responses for the same context

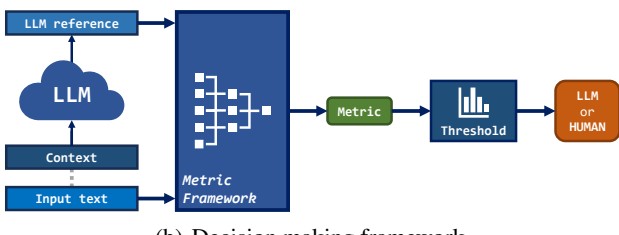

(b) Decision making framework

Figure 1: The metric-based detection framework and examples of justification

## 3.2 METRIC LEARNING WITH SAME-CONTEXT TRIPLETS

Among state-of-the-art metric learning approaches is the triplet loss training (Schroff et al., 2015). In brief, this method utilizes triplets of data instances, two having the same labels, and one with a different label, as training units. The training goal is for each instance in the triplets to undergo the same transformation to an embedding space where pairs of the same labels (positive pairs) are spatially closer than the pairs having different labels (negative pairs). Mathematically, the loss of one triplet is as $L = max(\|f(A) - f(P)\|^2 - \|f(A) - f(N)\|^2 + \alpha, 0)$ where $A$, $P$, and $N$ are instances in the triplet, $A$ and $P$ have the same label, and $N$ has a different one; $f(\cdot)$ represents the transformation to the embedding space which is parameterized with a deep neural network; and $\alpha$ is a margin hyper-parameter. $f(\cdot)$ is trained to minimize the total loss $\sum L$ across all training triplets.

However, modeling the problem as a classification task, i.e., having the model directly generate the labels "LLM" or "human" is demanding in terms of both data and computational resources. The reason is that the original triplet loss optimization has to model similarities among responses from very different topics which increases the complexity of the training and in turns makes the models more complex and/or difficult to converge. Therefore, instead of labeling all the responses "LLM" or "human" and let the model randomly sample positive and negative pairs, we *strictly constrain the instances in each triplet to come from the same contexts. Furthermore, the positive pairs, A and P, come from LLMs, and negative instances, N, are from human*. Overall, the ultimate goal of metric learning in identifying LLM generated contents is to increase similarity among LLM texts from similar contexts, and decrease that between LLM texts and human texts, under the same condition. *The similarity among human texts is not trained* in this framework as it is not necessary. Figure 2 illustrates the process of same-context triplet sampling and the training objective of the framework.

## 3.3 THE DETECTION FRAMEWORK ARCHITECTURE

We design the transformation neural network $f(\cdot)$ as a combination of an embedding model and a metric model. More specifically, the embedding language model vectorizes raw text data, and the metric model takes the vectorized texts then outputing their distance values. Decision making is performed based on the final output distances.

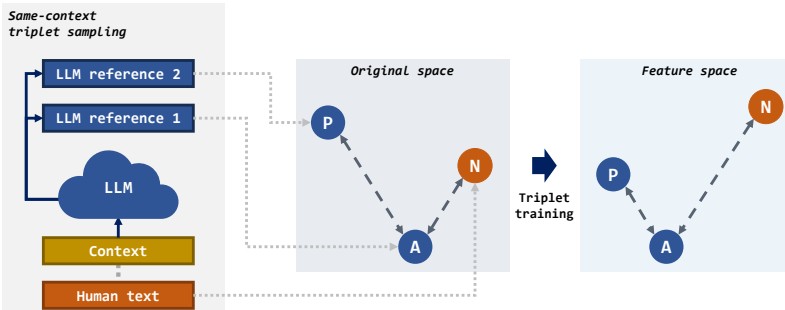

Figure 2: Same-context triplet sampling and training objective of the detection framework

To save computational resources as well as utilize knowledge from external domains, we use a public pretrained language model for the embedding task. At the moment, the selected embedding model is the pretrained MPNet version 2 (Song et al., 2020) available in the SentenceTransformer (Reimers & Gurevych, 2019) Python library. This model has 110 million parameters and was originally trained on 160GB of text data for various language tasks. It was then finetuned for the task of sentence embeddings on over a billion pairs of sentences and is current having the highest benchmark among pretrained models for this task. MPNet takes raw text data and outputs a 768-dimensional vectors. The outputs of MPNet are then fed to a deep metric model. As MPNet can operate at different levels of text granularity, i.e., at the token, sentence, or full-text, we design different metric architectures for each type. However, to keep the model complexity low, *we only consider embedding models at the sentence or full-text levels*.

The first metric model at the full-text level takes two input embeddings from MPNet and transforms them to higher-level vector representations. Then, the Euclidean distance of the two vectors is computed and used as the metric of the two original inputs. Architecture-wise, the full-text model consists of stacked residual blocks similar to that in the transformer model (Vaswani et al., 2017). More specifically, each residual block consists of two consecutive fully-connected layers, the first using Rectified Linear Unit (ReLU) activation, and the second layer, linear activation. The output of the linear layer is then added with the original block input and normalized to the final block output.

On the other hand, the sentence-level metric network receives an array of sentence embeddings from each input. Therefore, we design this model similar to the transformer architecture. Specifically, the model has two towers that take MPNet sentence embeddings from the input text and its LLM reference. Each set of embeddings undergoes a set of self-attention and feed-forward blocks. The outputs from the feed-forward blocks of the two towers are then merged in a cross-attention block where the LLM's outputs act as $Key$ and $Query$, and the input text, $Value$. Like in transformers, the block of self-attention, feed-forward, and cross-attention, is stackable as needed. Lastly, the output of the final cross-attention block is fed to a feed-forward architecture, flattened, and go through a single-output Sigmoid layer to generate the metric of the two original inputs. An illustration of the two metric frameworks is as in Figure 3.

To keep the problem complexity low, we freeze MPNet from any updates during training. Consequently, only the metric network component is trained using the same-context triplet training algorithm that is described previously. More specifically, each training instance is formed by a triplet of two LLM texts and one human from the same context. One of the LLM texts acts as the LLM reference, while the remaining one and the human text act as the second input to the metric network. Finally, a loss value is computed for each triplet of texts, summed over the training batch, and used to update parameters in the network using ADAM optimization (Kingma & Ba, 2014).

It should be noted that the triplet data is only required during the training phase. During the decision making process, only the context-text pair is needed. First, the context will be used as prompt to generate a LLM response. Then, both the text and the LLM response are input into the trained deep framework to generate their metric which is compared to a pre-selected threshold to determine whether the answer was generated by ChatGPT. In this paper, we tune the distance threshold in the validation data during model training.

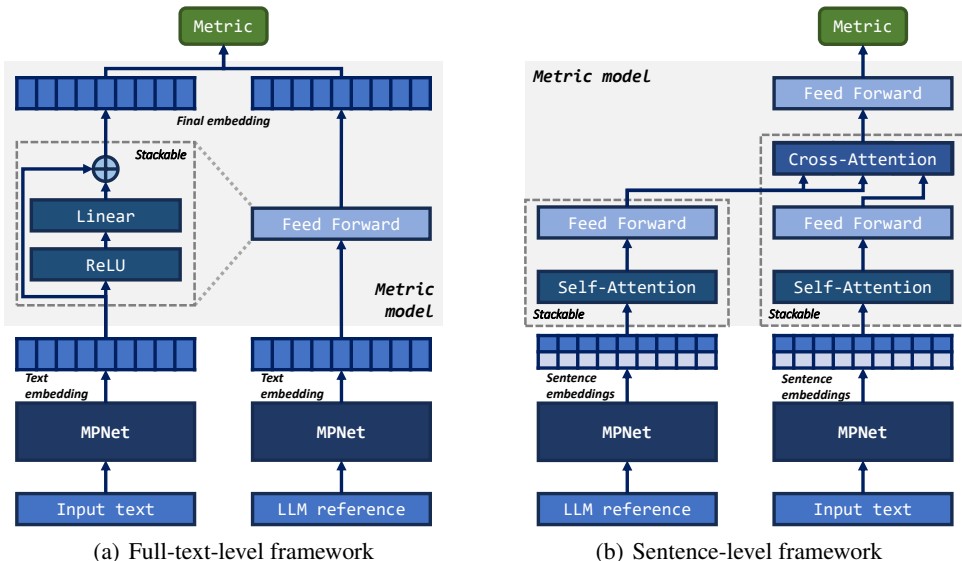

(a) Full-text-level framework      (b) Sentence-level framework

Figure 3: Metric neural network architectures

## 4 BENCHMARKING DATA

To our knowledge, text datasets consisting of contexts and texts from both human and LLM have yet existed. Therefore, we construct four of such collections for the purpose of developing and benchmarking our approach, as well as the general public to use. We focus on the most publicly accessible LLM currently which is GPT-3.5 TURBO through ChatGPT. The generation temperature is kept at $0.7$ which is the default value in the free version of ChatGPT. In terms of data sources, we utilize four public datasets, namely Natural Questions dataset (NQ) (Kwiatkowski et al., 2019), Stanford Question Answering data (SQUAD) (Rajpurkar et al., 2016), Scientific Questions dataset (SciQ) (Johannes Welbl, 2017), and the Wikipedia Scientific Glossary (Wikipedia, 2017).

First, NQ is a database of questions and annotated answers from Wikipedia pages on numerous topics. We use the simplified version of the data which consists of over $150,000$ questions, each comes with the complete content of the Wikipedia page that has the answer as well as the locations of the long answer and short answer as token indexes. For our usage, we extract the long answer for each question, remove all HTML tags, and filter to questions of which answers have at least five sentences and no more than 300 words. To generate ChatGPT answers, we use the prompt "*please answer the following question using at least five sentences*", follow by the actual questions in data. Each prompt is requested twice in two different API calls. The data is further filtered to exclude questions with ChatGPT responses of less than three sentences. The final data has $56,845$ instances.

Second, SQUAD is a dataset focusing on the question-answering task with over $100,000$ questions on contents from Wikipedia. This data is organized differently from the NQ data in that there are multiple questions for each paragraph in a Wikipedia article. To select the best question to represent a paragraph, first, all questions are sent to ChatGPT to obtain its responses. Then, we feed all ChatGPT responses as well as the original paragraphs through MPNet to obtain their embeddings. Next, the distances between each paragraph and their questions' LLM responses are calculated. Finally, the questions of which LLM answers yield the lowest distance to their corresponding paragraphs are selected. Like the NQ data, too short and too long texts are excluded from the result. Finally, in two different sessions, we feed the selected questions to ChatGPT to obtain the two LLM responses for this data. At the end, there are $18,813$ contexts and triplets of responses in this corpus.

Third, SciQ is a dataset consists of 13,679 scientific exam questions in topics like Physics, Chemistry, Biology, etc. All questions are multiple-choices, however, many come with a paragraph explaining the correct answer. For this dataset, we first filter to questions that have an explanation paragraph with at least three sentences. The questions are further constrained to those that are in the forms "what", "when", "where", "who", "which", and "how", to ensure the quality of the answer

explanations. Similar to the NQ data, we use the prompt "*please answer the following question using at least five sentences*" concatenated with the actual questions to obtain two responses from ChatGPT. Same questions are prompted in different API calls to make sure the two responses are unrelated. Another filter on removing ChatGPT responses of below three sentences is applied. The final SciQ data consists of $4,419$ instances of questions and their triplets of responses.

Finally, we extract data from the Wikipedia scientific glossary to build the last dataset (Wiki) on concepts in areas such as artificial intelligence, computer science, physics, biology, chemistry, economics, etc. First, the pages are cleaned of HTML tags then processed into the data format $\{concept, explanation\}$. The concepts are then filtered to those having explanations of at least two sentences. To generate ChatGPT responses, we use the prompt "*in at least three sentences, explain the concept of*", followed by the actual concepts in data. Like the previous datasets, responses for the same prompt are requested in different API calls to ensure their independence, and the resulted data is filtered to exclude questions with ChatGPT responses below three sentences. The final Wiki glossary data consists of $2,071$ triplet entries.

We conclude this section by emphasizing that the minimum answers' lengths in prompts cannot enforce that in responses from ChatGPT. For examples, prompts with "at least five sentences" may result in answers of shorter lengths but usually capped at five. This part of the prompt is to mainly to limit the chances of one-sentence answers from ChatGPT. Furthermore, the human responses in the Wiki data are shorter than in the NQ, SQUAD, and SciQ datasets, on average. Therefore, the minimum answer lengths for prompts from this data is smaller (three for Wiki vs. five for the others). We aim for the response lengths to be approximately similar between human and ChatGPT.

## 5 EXPERIMENT STUDY

We test our models in three experiment settings: same-corpus, same-corpus with paraphraser, and out-of-corpus. In the two same-corpus settings, the datasets are tested independently. Each is split into 80% training, 10% validation, and 10% testing. In the same-corpus with paraphraser experiment, we feed all responses from both human and ChatGPT through the Parrot model (Damodaran, 2021) which is a finetuned T5 language model (Raffel et al., 2020) for text paraphrasing. Then, the original version and paraphrased version of the training and validation data are merged, whereas the testing data only comes from the paraphraser. Lastly, in the out-of-corpus setting, we perform three tests: 1) training and validation with NQ, testing with SQUAD; 2) training and validation with SciQ, testing with Wiki; and 3) training and validation with NQ merged SciQ, and testing with SQUAD merged Wiki. In all cases, the ratio of training-validation is $90\% - 10\%$.

For the full-text-level framework (denoted *full-text*), we finetune the number of feed-forward blocks in the metric network in $[1, 12]$. The $\alpha$ margin in triplet loss is finetuned within the values of $\{0.01, 0.02, 0.05, 0.1, 0.25, 0.5\}$. The final architecture consists of three residual blocks, with $\alpha = 0.1$, and is trained in 50 epochs using ADAM optimization learning rate of $10^{-5}$ with a batch size of $2048$. On the other hand, due to its more complex architecture, the sentence-level framework (denoted *sentence*) is finetuned with feed-forward blocks in $[1, 3]$ and attention blocks in $[1, 3]$. $\alpha$ is finetuned in similar range with the full-text-level framework and is $0.25$ at the end. The final sentence metric model consists of one attention block and one feed-forward block, is trained in 100 epochs with learning rate of $5 * 10^{-5}$ epochs, and batch sizes of $1024$. In all architectures, the number of neurons in all blocks' hidden layers are fixed at 768 which is the dimensionality of embeddings output by the pretrained MPNet.

To evaluate our model, we use two metrics, **triplet accuracy**, and **F1 score**. Triplet accuracy is computed as the ratio of triplets where the output embeddings of negative pairs (human response and ChatGPT response) do have longer distances than the that of positive pairs (two ChatGPT responses). We utilized this metric since it directly reflects the quality of the triplet training process. To measure the actual detection quality, we utilize F1 score. Unlike during training when data comes as triplets, the prediction phase uses data as pairs: a given answer and its corresponding ChatGPT response. Therefore, each triplet is broken down into two pairs to form the prediction data. As discussed, to make prediction, a distance threshold is selected. If a response's distance to its corresponding ChatGPT counterpart is above the threshold, the response is labeled as human generated, otherwise, it is ChatGPT generated. We finetune the distance threshold separately each test run by optimize thing F1 in the validation data. This value is typically around $0.3$.

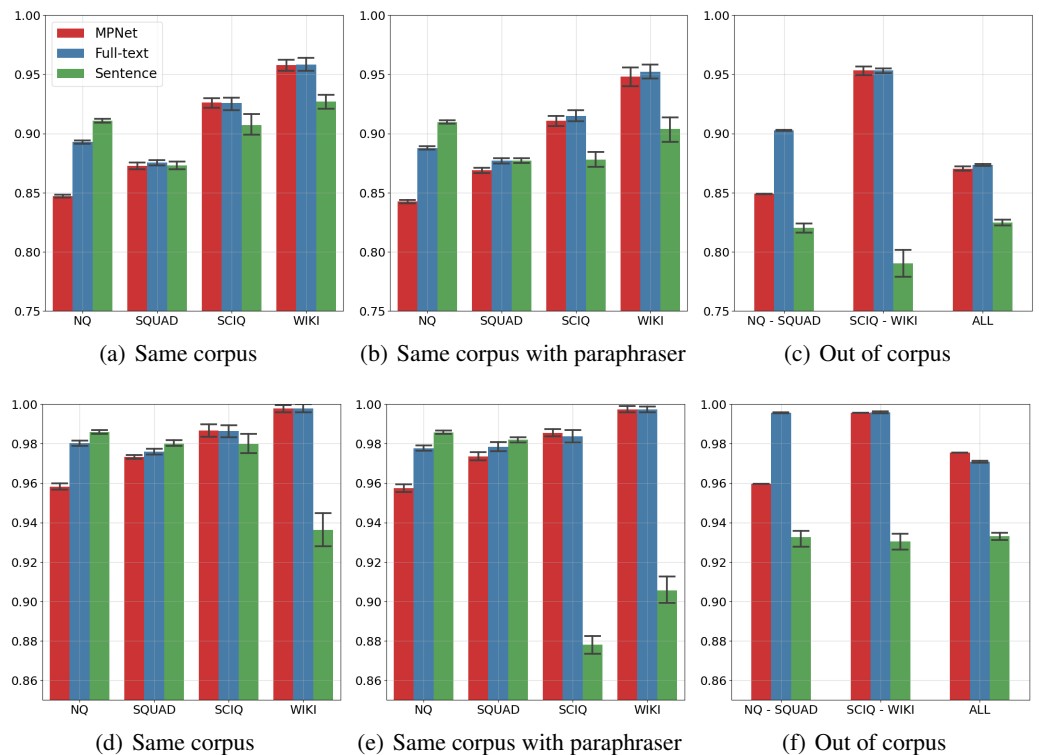

Figure 4: Models' F1 (a)(b)(c) and triplet accuracy (d)(e)(f) in experiment study

As a baseline, we apply the distance threshold approach on the embedding generated by the standalone pretrained MPNet (denoted *MPNet*). We also attempted to train classifiers on the responses' MPNet embeddings, however, ***these supervised classifiers failed to converge*** even at much higher numbers of parameters compared to the triplet trained models (up to 20 layers - 12 million parameters). Furthermore, we deem finetuning large language models by updating all weights too computationally expensive which is then excluded in our experiments. For illustration, using a T4 graphical unit, ***the full-text model uses*** $8$ ***seconds for one epoch*** in the NQ data (about $51,250$ triplets in training and validation) at batch size $2,048$, and ***the sentence model*** needs about ***three minutes***, batch size $1,024$. In contrast, ***one epoch of finetuning DistilBERT*** (Sanh et al., 2019) on the NQ data split takes ***70 minutes***, and ***RoBERTa*** (Liu et al., 2019), ***125 minutes***, both at the highest batch size allowable of $16$. Lastly, the probabilistic models and watermarking are not considered as they need accesses to internal computations of LLMs and do not fit our accessibility criterion.

To measure performance, all models, full-text, sentence, and base, are tested in 10 runs. Hyperparameters are fixed across runs of the same experiment settings, except for the detection distance threshold which is tuned separately by runs using validation F1. In terms of result, the model performances are illustrated as bar charts in Figure 4. Figure 4 (a)(b)(c) show F1 scores of the models in each dataset for the same-corpus, same-corpus with paraphraser, and out-of-corpus experiments, respectively. Figure 4 (d)(e)(f) present the triplet accuracy of the models in the same order. First, we can observe that, the baseline MPNet models actually perform very well in all experiments. It achieves over $96\%$ triplet accuracy and $0.84 - 0.96$ F1 in all experiments. Next, the full-text model outperforms the base model in almost all experiments, especially in the NQ and SQUAD data. In the SciQ and Wiki data, the two models are relatively similar in both F1 and triplet accuracy. Finally, the sentence model is the best one in the NQ and SQUAD data in the same-corpus settings, while being the worst in the rest of the experiments. This result is explainable as its higher complexity leading to more data required to generalize well, especially to out-of-corpus data. More specifically, in the same corpus settings, with the NQ data having sufficient size, the sentence model largely outperforms others. Its performance then drops to similar to the full-text model in SQUAD, and becomes the lowest in SciQ and Wiki. In the out-of-corpus setting, it seem that none of the tested training data is large enough for the sentence model to generalize well.

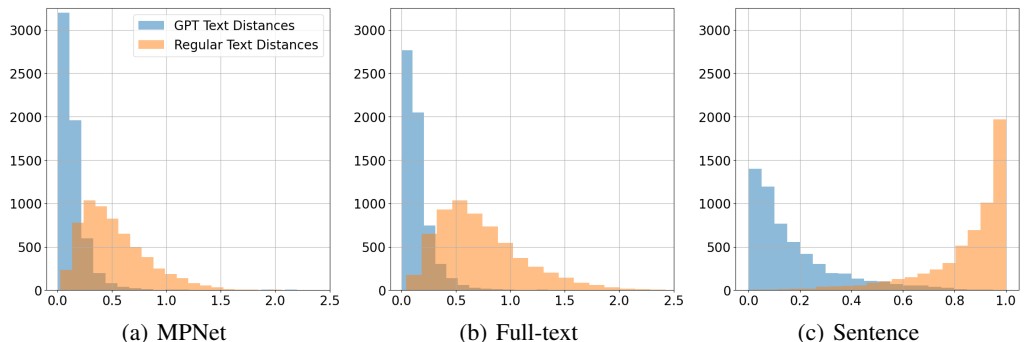

(a) MPNet  (b) Full-text  (c) Sentence

Figure 5: Histograms of texts' distances in NQ test data

Finally, we investigate the distributions of trained distance metrics in Figure 5. The histograms are constructed from the testing portion of the NQ data in the same-corpus setting. Again, we can see that the base models are powerful by themselves and can form two clear distributions of ChatGPT-ChatGPT and ChatGPT-human texts. In the full-text model, distances between ChatGPT responses are further reduced, whereas that of ChatGPT and human responses are stretched further to the right. Its distribution shape still resembles the base model, overall. The sentence model, on the other hand, transform the distance distribution totally – two distance clusters are clearly presenting with minimal overlapping. This shows its potentials when the training corpus is large enough to maintain its generalization capability. Regardless, based on the experiment results, we consider the full-text model as the most balance among the three in terms of performance and complexity.

## 6  CONCLUSION

The recent breakthrough in large language models, while offering tremendous potentials to society, also brought forefront the needs of effective methods to identify if contents are generated by human or artificial intelligence. However, there are currently two gaps in this research area, 1) identification technologies developed before 2022 require expensive computational resources and also have not been tested with the current flagship LLMs; and 2) current approaches, while offer very good detection accuracy, rely on internal computations of the LLMs. In education, science, or any areas where original contents are critical to have accurate assessments on teaching and learning quality, this issue only becomes more critical.

With such motivation, in this paper, we focus on the task of identifying whether texts are originated by human or LLM with a light-weighted and accessible approach. First, we observe that responses from LLM usually contains repeated patterns for the same prompts. Exploiting that observation, our detection framework is trained to signify the similarities among LLM responses while boosting their dissimilarities to human-generated ones. More specifically, the framework consists of an embedding component and metric component. The embedding component is pretrained to output vector representations for raw text data. The metric neural network then further take the embedding vectors to generate their distances. Architecture-wise, we propose two metric models, one at the full-text level, and one at the sentence level. The full-text model consists of stacked feed-forward blocks, whereas the sentence model follows the transformer architecture. Both models are trained using the same-context triplet algorithm designed for LLM text detection. Experiments show that our best models obtain F1 in between $0.85 - 0.95$ in multiple settings, while can converge very fast, typically 50-100 epochs each of which is at tens of seconds on training data of $51,250$ triplets.

For future works, we will explore the following directions. First, we will explore more architectures for the metric components, as they are the core of this detection paradigm. Second, we will develop methods that can effectively reconstruct contexts from any texts, as this information is not always available. Besides using questions, some works have utilize a start portion of the texts themselves, which limit the use cases to longer inputs. Finally, we will adapt this work to the general cases such as modeling longer texts in the form of essays or documents, or where the generative LLMs are not known at decision-making times.

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
