# OpenReview forum: "Metric Learning for Detection of Large Language Model Generated Texts"
_ICLR.cc/2024/Conference — Submitted to ICLR 2024_

### Official Review · Reviewer_8Zoy · 2023-10-31

**Soundness:** 3 good
**Presentation:** 3 good
**Contribution:** 3 good
**Rating:** 6
**Confidence:** 4

**Summary:**

The paper proposed a solution to detecting texts generated by LLMs. It emphasized on computational costs, accessibility, and performance.

Specifically, the authors propose a metric-based detection framework that evaluates the similarity between a given text and an equivalent example generated by LLMs to determine the text's origination. The framework includes a text embedding model and a metric model, with a focus on designing the metric component.

The authors also introduce four datasets with over 85,000 prompts and triplets of responses for benchmarking, showing that their best architectures maintain F1 scores between 0.87 to 0.95 across various settings.

**Strengths:**

* The approach itself is new. Being able to detect this without access to the internal structure is very valuable.
* The data sets will be very useful for any downstream tasks.
* The experimental results are impressive.
* The focus on computational cost and accessibility is especially relevant.

**Weaknesses:**

* This approach still relies heavily on the LLM themselves.
* The scope of the study is rather small, focusing only on short responses.

**Questions:**

* Could you elaborate on the different types/variations of prompts?

---

> ### Author Response · Authors · 2023-11-23
>
> Dear reviewer,
>
> Thank you very much for your high regard on our paper. To address your concerns:
> 1. We utilize the LLMs themselves as a method to reduce computational costs from the detectors. Having references from the generative LLMs, the detectors can focus more on learning a good metric while still maintaining reasonable complexity.
> 2. We are expanding this works to longer text data.
> 3. The prompts are mainly questions in the forms what/who/when/where/which/how.
>
> Thank you very much again.

---

### Official Review · Reviewer_CEgw · 2023-10-31

**Soundness:** 2 fair
**Presentation:** 1 poor
**Contribution:** 2 fair
**Rating:** 3
**Confidence:** 5

**Summary:**

The authors propose a metric-learning based approach for the detection of LLM-generated text given a known context. The authors propose two metric-based neural architectures trained with triplet loss, one based on embedding the full-text, and another based on embedding individual sentences. The detection decision is then based on thresholding the distance between an input-text and an LLM generated text.

**Strengths:**

* Approach does not require knowing the logits of the LLM or modifying its logits in any way as with watermarking.
* Focuses on relatively short pieces of text (3-5) sentences.
* Authors created and will release a set of four datasets totaling over 85,000 generations on a wide variety of topics.

**Weaknesses:**

* There are many typographic errors spread throughout the manuscript. This made the paper overly difficult to understand.
* The proposed approach requires knowing the context that may have been given to an LLM to generate the text. This seems like too big of a restriction, as it is often the case that we don’t know how the LLM was prompted. Suggestion:
  * Instead of requiring the context to generate a response by ChatGPT, a set of known ChatGPT generations on arbitrary generations could’ve been kept aside. The detector would then take the distance between the input text and the set of known ChatGPT generations.
  * The above could be compared to the case where one does know the context of the generation.
* The only LLM considered is ChatGPT. It would’ve been interesting to include a broader set of LLMs and explore the robustness of the approach to unseen LLM. Suggested Models: Llamav1, Llamav2, OPT, GPT-2, GPT-4, Cohere, Dollyv2, etc.
* In the “same corpus with paraphraser” experiments the model was trained on the paraphrased LLM detections. This means that all the testing data is still in-domain, and hence it may defeat the purpose of the experiment. Suggestions:
  * Use a different paraphraser on the testing data.
  * Do not train on the paraphrased text.
* It is unclear whether the sentence framework out-performs the full-text framework simply because it has more parameters. Suggestion: To experiment whether this is true, a best-effort could be made at matching the number of parameters between both frameworks by either increasing the number in the case of the full-text framework or decreasing it in the case of the sentence framework.
* Instead of evaluating the F1 score on the best threshold found in the validation set, it would’ve been better to just plot the ROC curves and show the median and 90th / 10th percentiles. This would give a better understanding of how the detector performs across varying ranges of FPR.
* There are more up-to-date metric-learning losses that could’ve been explored. Examples:
  * Supervised Contrastive Loss: https://arxiv.org/abs/2004.11362
  * InfoNCE: https://papers.nips.cc/paper_files/paper/2016/file/6b180037abbebea991d8b1232f8a8ca9-Paper.pdf
  * Tuplet Margin Loss: https://openaccess.thecvf.com/content_ICCV_2019/papers/Yu_Deep_Metric_Learning_With_Tuplet_Margin_Loss_ICCV_2019_paper.pdf
* Not enough baselines compared against. Some recommendations include:
  * OpenAI Detector: https://huggingface.co/roberta-base-openai-detector
  * LLMDet: https://github.com/TrustedLLM/LLMDet
  * A RoBERTa detector fine-tuned on the context and LLM-generation / human-answer for each dataset.

**Questions:**

* In S5, the training time of various methods is brought up as a reason to exclude certain baselines from consideration. However, this is an offline cost, which is not relevant at inference/test-time, so it's not clear why it's relevant. How do various methods compare at inference time?

* In practice, the robustness of machine-text detectors is important for many applications of machine-text detectors. For example, the cost of false positives is high when making false accusations of plagiarism. As a result, it is important to measure the robustness and calibration of such detectors. How does the proposed method fare when faces with various distribution shifts relative to the training data? (new LLMs, new domains, new decoding methods, etc.) Is it more robust than simple alternatives, such as simple supervised classifiers?

---

> ### Author Response · Authors · 2023-11-23
>
> Dear reviewer,
>
> Thank you very much for your feedback. We want to address your comments as follows.
> 1. We are developing another model to reconstruct prompts from arbitrary texts. These models can be as simple as a finetune LLM to pose a question that capture the topic of a text.
> 2. At the time of model training and evaluation of models in this paper, GPT-3.5-TURBO/ChatGPT was the most popular and advance. We will incorporate the newer models into this work.
> 3. While the models are trained on paraphrased texts, the question-answer data is not the same between training and testing. Furthermore, the test data only comprises of paraphrased texts. Therefore, we believe this experiment makes sense as-is. Regardless, we will follow your suggestion.
> 4. Contrastive loss is fairly older than triplet loss. We will consider the other two.
> 5. We eliminate a lot of detectors from comparison due to their complexity. We are aiming at small models that can be maintained in consumer computers.
>
> Finally, to answer your questions:
> 1. While computational costs of supervised classifiers are indeed cheaper at inference time, they still require a big enough system to host/maintain. For example, ROBERTA at 1.5 billion parameters is definitely not easy for a regular consumer computer to sustain. As we really want to aim at cheap models, complexity in inference is also of interests.
> 2. Expanding this model to general/new LLMs and domains is one future direction of this project.
>
> We will incorporate your suggestion into the next iteration of this paper.
>
> Thank you very much again for your valuable feedback.

---

### Official Review · Reviewer_7NGJ · 2023-11-01

**Soundness:** 2 fair
**Presentation:** 3 good
**Contribution:** 2 fair
**Rating:** 3
**Confidence:** 4

**Summary:**

This paper studies the problem of detecting texts generated by Large Language Models(LLMs). The authors propose a method that involves assessing the similarity between a given text and a comparable LLM-generated example to determine the source of the text. The authors conduct experiments on a novel, metric-based detection paradigm, utilizing four extensive datasets comprising over 85,000 prompts and response triplets, to evaluate the effectiveness of their approach in distinguishing between human-written and LLM-generated texts across various corpora and contexts, including paraphrasing scenarios. Experimental results show the effectiveness of the proposed method.

**Strengths:**

- The paper is well-organized and easy to follow.

**Weaknesses:**

- The paper focuses exclusively on short text responses, with an average length of five sentences and no less than three. This limitation could hinder the generalizability and applicability of the proposed approach, as longer texts or documents might exhibit different characteristics and patterns that are not captured by the model trained on shorter responses.

- Lack of convincing baselines. The authors merely use the distance threshold approach on the embedding generated by the standalone pretrained MPNet as the baseline.

- The authors not evaluate the proposed method on the benchmark such as Human ChatGPT Comparison Corpus (HC3) [4].

- There is a lack of previous studies in the literature.

[1] Gehrmann, Sebastian, Hendrik Strobelt, and Alexander Rush. "GLTR: Statistical Detection and Visualization of Generated Text." Proceedings of the 57th Annual Meeting of the Association for Computational Linguistics: System Demonstrations. Association for Computational Linguistics, 2019.

[2] Clark, Elizabeth, et al. "All That’s ‘Human’Is Not Gold: Evaluating Human Evaluation of Generated Text." Proceedings of the 59th Annual Meeting of the Association for Computational Linguistics and the 11th International Joint Conference on Natural Language Processing (Volume 1: Long Papers). 2021.

[3] Sadasivan, V. S., Kumar, A., Balasubramanian, S., Wang, W., & Feizi, S. (2023). Can ai-generated text be reliably detected?. arXiv preprint arXiv:2303.11156.

[4] Guo, B., Zhang, X., Wang, Z., Jiang, M., Nie, J., Ding, Y., ... & Wu, Y. (2023). How close is chatgpt to human experts? comparison corpus, evaluation, and detection. arXiv preprint arXiv:2301.07597.

**Questions:**

Please refer to the "Weaknesses" section.

---

> ### Author Response · Authors · 2023-11-23
>
> Dear reviewer,
>
> Thank you very much for you feedback. We want to address your second point regarding the baseline. As we eliminate supervised classifiers due to their complexity, and probabilistic models as they require access to output logits, the choice of baselines is limited. We will take your suggestion and incorporate more literature into the next iteration of this paper.
>
> Thank you again.

---

### Official Review · Reviewer_X5RJ · 2023-11-03

**Soundness:** 1 poor
**Presentation:** 3 good
**Contribution:** 2 fair
**Rating:** 3
**Confidence:** 3

**Summary:**

This paper proposes a neural classifier that detects LLM-generated texts from
human-generated texts. Instead of directly relying internals of LLMs, the
proposed classifier leverages triplet samples of the same meanings, and builds
a neural classifier on these triplets that uses metric learning internally.
Experimental results indicate that the proposed classifier works good (but not
shown better than baselines) on several datasets, and induced metrics are
significantly different between LLM-generated texts and human-generated texts.

**Strengths:**

While the rationale and the proposed approach is decent, the crucial drawback
of this paper is that it is not compared with possible baselines. The basic
idea that the classifier should base on the sentences of the same semantic
content is of course good, but such a classifier can be built just from a set
of pairs of texts, not triplets. Also, that central idea of using the
same-meaning sentence itself should be validated empirically: how about the
performance if we simply build a classifier on two sets of texts, one from LLM
and one of humans?

Without such empirical investigations, this paper should not be accepted as a
machine learning conference paper.

**Weaknesses:**

See above.

**Questions:**

Nothing.

---

> ### Author Response · Authors · 2023-11-23
>
> Dear reviewer,
>
> Thank you very much for you feedback. We want to clarify that the reason triplet training was selected is because it had advantages over pairwise training. First, the performance of triplet loss was proved to be very good in multiple research. Second, pairwise training like contrastive loss could be more extreme in that the metric of same-label pairs are learned to be 1, and that of different-label pairs is learned to be 0, which could pose a more difficult training problem.
>
> We will take your suggestion into consideration and include pairwise models in the next iteration of this paper.
>
> Thank you again.

---

### Meta-Review · Area_Chair_bv55 · 2023-12-10

**Metareview:**

This paper proposes a method to detect whether a text is model generated, using similarity between the text and generated text, based on learned metrics. To this end, the authors create four extensive datasets comprising over 85,000 prompts and response triplets.

Strengths: The paper deals with an important AI safety and trust problem. The method tries to approach this problem without access to model probabilities / logits, which is a very practical setting.

Weaknesses: While the paper presents empirical results, the lack of sufficient baseline / benchmark comparisons as well as a somewhat limited exploration of settings (shorter sequences, limited model families), came across as a major limitation of the work.

**Justification For Why Not Higher Score:**

The empirical results are not presented with respect to existing baselines. Different model choices need to be explored to validate the efficacy of the approach.

**Justification For Why Not Lower Score:**

N/A

---

### Decision · Program_Chairs · 2024-01-16

Reject